Characteristics of solar radiation at Xiaotang, in the northern marginal zone of the Taklimakan Desert

Jin Lili 1 2 3
Zhou Sasa 2 3
He Qing qinghe@idm.cn 2 3
Abbas Alim 2 3
1 Department of Atmospheric Sciences, Yunnan University , Kunming , China
2 Institute of Desert Meteorology, China Meteorological Administration , Urumqi , China
3 Taklimakan Desert of Xinjiang, Desert Meteorology, National Observation and Research Station , Urumqi , China
Wang Jingzhe
Electronic publication date: 2021 Nov 12
Publication date: 2021
Volume: 9
Electronic Location ID: e12373
Received 2021 Jun 7; Accepted 2021 Oct 3
Copyright: ©2021 Jin et al.
Copyright year: 2021
Copyright holder: Jin et al.
License: This is an open access article distributed under the terms of the Creative Commons Attribution License, which permits unrestricted use, distribution, reproduction and adaptation in any medium and for any purpose provided that it is properly attributed. For attribution, the original author(s), title, publication source (PeerJ) and either DOI or URL of the article must be cited.
License URL: https://creativecommons.org/licenses/by/4.0/

Keywords: Total radiation, Direct radiation, Scattered radiation, Taklimakan Desert

Funding: National Natural Science Foundation of China 41830968 42030612 This work was financially supported by the National Natural Science Foundation of China (41830968,42030612). The funders had no role in study design, data collection and analysis, decision to publish, or preparation of the manuscript.

==============================
The characteristics of solar radiation and the influence of sand and dust on solar radiation in the northern margin of Taklimakan Desert were analyzed using radiation observation data from 2018. The results showed that the annual total radiation, direct radiation, and scattered radiation at Xiaotang were 5,781.8, 2,337.9, and 3,323.8 MJ m−2, respectively. The maximum monthly total radiation, direct radiation, and scattered radiation were observed in July (679.8 MJ m−2), August (317.3 MJ m−2), and May (455.7 MJ m−2), respectively. The aerosol optical depth corresponded well with the scattered radiation, and the maximum value was in May. Further analysis showed a significant correlation between the total radiation and solar height angle under different weather conditions. Under the same solar height angle, total radiation was higher during clear days but lower on sandstorm days. Calculation of atmospheric transmittance showed that the average atmospheric transmittance on a clear day was 0.67; on sand-and-dust days, it was 0.46. When the atmospheric transmittance was 0.5, the increase in scattering radiation by aerosol in the air began to decrease. Probability analysis of radiation indicated the following probabilities of total radiation <500 W m−2 occurring on clear, floating-dust, blowing-sand, and sandstorm days: 67.1%, 76.3%, 76.1%, and 91.8%, respectively. Dust had the greatest influence on direct radiation; the probabilities of direct radiation <200 W m−2occurring on clear, floating-dust, blowing-sand, and sandstorm days were 44.5%, 93.5%, 91.3%, and 100%, respectively, whereas those of scattered radiation <600 W m−2were 100%, 99.1%, 98.1%, and 100%, respectively. Therefore, the presence of dust in the air will reduce scattered radiation.

Introduction

Solar energy is the most fundamental renewable energy source on the earth’s surface, and global solar radiation plays an important role in a wide range of applications, such as in the areas of meteorology and hydrology (Almorox, Bocco & Willington, 2013). Solar radiation is a primary factor in several applications, such as solar energy systems, architecture, agriculture, and irrigation (Almorox & Hontoria, 2004). The solar radiation that reaches the ground is divided into two parts: solar direct radiation and scattered radiation. The former is directly projected onto the ground in the form of parallel rays, and the latter is projected from the sky to the ground after scattering. The sum of these two radiations is called total radiation. The solar radiation reaching the ground is mainly affected by astronomical factors and the earth’s atmosphere (Bullrich, 1964).

Many studies state that the total surface radiation and direct radiation have been decreasing in most parts of the world in the 20th century, which may be due to the increase in the concentration of suspended particles in the atmosphere (Gilgen, Wild & Ohmura, 1998; Wild et al., 2005; Wild, 2012). Che et al. (2005) evaluated China’s solar radiation data from 1961 to 2000; they found that the total radiation had decreased significantly (4.5 W m−2 per decade), but the diffuse fraction had increased (1.73% per decade), and an increase in aerosol is partly the reason for the decrease in the observed solar radiation.

Dust is a common type of aerosol found in deserts (Smirnov et al., 2002; Masmoudi et al., 2003; Oh et al., 2003). It is the most important component of aerosols and accounts for approximately one-third of the total amount of naturally generated aerosols worldwide (Miller, Tegen & Perlwitz, 2004; Han et al., 2008), which equates to approximately 1,000–4,000 Tg per annum (Huneeus et al., 2011; Jemmett-Smith et al., 2015). Dust aerosols can affect weather and global and regional climate change (Huang et al., 2014; Othman et al., 2010; Kang et al., 2011). They also disturb the radiation–energy balance of the earth–atmosphere system and cause severe biological and ecological impacts that affect human health (Othman et al., 2010; Falkowski, Barber & Smetacek, 1998; Goudie, 2014). Changes in the number and size of dust particles suspended in the air change their optical properties. According to Xin et al. (2003), on dusty days, the direct radiation in the presence of dust aerosols attenuates by 38% on average. During sandstorms in the Sahara Desert, the direct interaction of dust with radiation causes an additional reduction of 40–80 W m−2 in the incoming shortwave radiation. The strong radiative force associated with dust causes a reduction in the surface temperature in the order of −0.2 to −0.5 K in most parts of France, Germany, and Italy during dust events (Bangert et al., 2012; Slingo et al., 2006). The use of radiation models to simulate radiative fluxes underestimates the observed absorption of solar radiation in dusty atmospheres (Davidi et al., 2012). The dust in the atmosphere greatly disturbs the total radiation balance of the underlying surface while inducing general warming of the underlying surface–atmosphere system owing to a decrease in the system albedo over arid zones (Sokolik & Golitsyn, 1993a; Sokolik & Golitsyn, 1993b). Mani & Chacko (1980) discussed the contribution of dust to the attenuation of solar radiation in the Rajasthan Desert by the scattering and absorption of radiation. In summer, the dust aerosols attenuate solar radiation by up to 40%. Using MODIS satellite observation data, Hatzianastassiou et al. (2014) found that aerosols in the desert regions of Asia and Africa strongly absorb 15–55 W m−2 of incident solar radiation, with the maximum absorption occurring in the Sahara.

In China, strong sand and dust events mainly occur in the Taklimakan Desert and the eastern and northern parts of northern China (Zhou & Wang, 2002). Previous studies on radiation in the hinterland of the Taklimakan Desert have examined the impact of sand and dust on radiation (Meng & Li, 2019; Fei, Xia & Che, 2014); however, there are limited studies on the impact of sand and dust on radiation in the margin of the Taklimakan Desert. Huang et al. (2020) attempted to improve the accuracy of the simulation analysis of clear-sky surface shortwave radiation using the CERES SSF dataset, and aerosol optical depth (AOD) plays a crucial role in deciding the accuracy of simulation analysis. Jin et al. (2011) conducted UAV experiments and found that particles size, concentrations, and daily variations in the concentration of PM were different in the Xiaotang and Tazhong Station. On clean days, the mean concentrations of PM1.0, PM2.5, and PM10 were 3.6, 6.9, and 18.9 µg m−3 in the Xiaotang area and 6.5, 13.2, and 26.8 µg mg m−3 in the Tazhong area, respectively. There are differences in natural environment and altitude between the hinterland and the margin of Taklimakan Desert. Using radiation observation data from the Xiaotang land–atmosphere interaction observatory and experiment station (40.8°N, 84.3°E), we can further understand the characteristics of radiation changes in the desert margin area during the dust weather in 2018. The findings will be of great significance for understanding climate change and ecological protection in the arid region of northwestern China, and they will provide basic data for studying the radiation balance of oasis–desert transition zones and the effects of dust aerosols on radiation.

Overview of Study Area, Data, and Methods

Study area

Xiaotang is located at the northern margin of the Taklimakan Desert, northwestern China (Fig. 1). This area lies in the desert–oasis transition zone on the northern margin of the Taklimakan Desert. The underlying surface is a relatively flat, bare, wind-eroded ancient riverbed. It has a warm temperate desert climate that is extremely arid and has a high potential for evaporation. The annual average temperature, precipitation, and wind speed are 10 °C, 15.2 mm, and 2.5 m s−1, respectively. The frequency of sandy and dusty weather is extremely high. Sandstorms and dusty devils occur in all seasons. Surface soils mainly comprise fine sand (125–250 µm) and very fine sand (62.5–125 µm) (Ma et al., 2020).

Figure 1 Geographical location diagram of Xiaotang, Taklimakan Desert.

The observation site is in the northern margin of Taklimakan Desert. The Xiaotang land–air interaction observatory station (40.8°N, 84.3°E; altitude = 932 m) is on the south bank of the ancient riverbed, ∼2 km north of a Populus euphratica forest that is minimally affected by human activities. According to the meteorological data from the Xiaotang observatory station, the average temperature of the northern margin of Taklimakan Desert in 2018 was 11.3 °C, the highest temperature was 39.4 °C (in summer), and the lowest temperature was −40. 1 °C (in winter) (Fig. 2). In 2018, the annual average wind speed was 1.6 m s−1, and the annual relative humidity was 34.95%.

Figure 2 Monthly changes in wind speed, temperature, and humidity at the Xiaotang area in 2018.

Observation instrument

The Xiaotang land–atmosphere interaction observatory and experiment station used internationally recognized radiation detection sensors (Table 1). The measured parameters included total, direct, and scattered radiations. The model of A CR3000 micrologger (Campbell, United States) was used as the radiation data collector. The acquisition frequency was 1 s, and the output data were obtained at 1 s, 1 min, 30 min, and 1 h. In this paper, we used hourly data for our analysis. The clock of the collector adopted the local real solar time, which later Beijing time by 2 h 22 min 48 s.

Table 1 Radiation detection instruments.

Sensor	Model	Installation height(m)	Producing country and manufacturer	Technical index	
Total radiation	SR20	1.55	Netherlands Kipp&Zonen	Spectral range: 300–2,800 nm; Sensitivity: 15 µV W−1; Zero drift: <5W m−2; Operating temperature: −40 °C–80 °C	
Direct radiation	DR20	1.35	Netherlands Kipp&Zonen	Spectral range: 200–4,000 nm; Sensitivity: 7–15 µV W−1; Temperature drift: <1 W m−2; Operating temperature: −40 °C–80 °C	
Scattering radiation	SR20	1.55	Netherlands Kipp&Zonen	Spectral range: 300–2,800 nm; Sensitivity: 15 µV W−1; Zero drift: <5W m−2; Operating temperature: −40 °C–80 °C	

According to the specifications, the radiometer was inspected and maintained every day, and the radiometer was wiped before sunrise. Some false data appeared due to instrument system errors, instrument failure, etc. during data transmission and recording. The following data must be corrected or eliminated: the output flux value is NAN.

Data and methods

The satellite data were obtained from the CERES_SYN1deg_Ed4A daily product data provided by the Atmospheric Science Data Center at NASA Langley Research Center. The accuracy of the CERES sensor for the atmospheric AOD measurements was 1°. Aerosol optical thicknesses were obtained using an aerosol transport model MATCH (Collins et al., 2001) that assimilated and spatially as well as temporally interpolated MODIS aerosol optical thickness. Additionally, MATCH provided aerosol types. The atmospheric AOD data were downloaded from the following link: http://ceres.larc.nasa.gov/.

The total, direct, scattered radiations’ data and ground meteorological observation data from the Xiaotang land–atmosphere interaction observatory and experiment station from January 1, 2018 to December 31, 2018 were used in this study. In this study, the months of March–May, June–August, September–November, and December–February are considered as spring, summer, autumn, and winter, respectively.

Kipp & Zonen’s A2P automatic tracker direct radiation meter was used to measure the direct solar radiation on the vertical plane; however, the direct solar radiation on the horizontal plane was required to analyze the direct radiation. Therefore, the direct radiation measured by the instrument was converted using the following formula (Mamtimin et al., 2014): (1) Rb=Rb sinh,

where Rb is the direct horizontal radiation, Rb ’ is the direct normal radiation, and h is the solar height angle.

In (Eq. 1), the solar height angle was calculated using the following equation: (2) sinh= sinφsinδ+ cosφcosδcost,

where φ (radian) is the local latitude, δ (radian) is the solar declination, and t is the hour angle.

The ratio of total radiation to total astronomical radiation is equivalent to the transmittance of the whole atmosphere. This parameter is called atmospheric transmittance in this paper (Tian et al., 2018).

The atmospheric transparency coefficient is an essential parameter for characterizing the degree of atmospheric turbidity. Therefore, the variation of the atmospheric transparency coefficient was used to discuss the influence of dust aerosols on direct solar radiation. The atmospheric transparency coefficient was calculated according to Eqs. (3) and (4). (3) S=S0Pm

(4) P=SS01/m

where S is direct radiation, S0 is the solar constant, and m is the atmospheric mass. We obtained the value of S when m = 2, and calculated P 2 according to (Eq. 5). (5) P2=m/2Pm1−12mPm,

where P2 isthe atmospheric transparency when the relative atmospheric mass is corrected to be 2 (that is, the solar altitude angle = 90°).

The characteristics of total, direct, and scattered radiations at Xiaotang during different seasons and sandy and dusty weathers were analyzed. Five typical days were selected for each clear, floating-dust, blowing-sand, and sandstorm weathers from spring to summer, when sandy and dusty days were more frequent. However, owing to limitations in the weather phenomena, four sunny days were selected for spring and summer and two floating-dust days were selected for summer. The daily changes in the sand and dust at Xiaotang were analyzed. Because dusty weather at Xiaotang was mainly observed during spring and summer, autumn and winter were not analyzed here.

A total cloudiness of <20% and no weather phenomenon indicated a clear day. When the dust and fine sand floated evenly in the air such that the horizontal visibility was <10 km, indicated a floating-dust day. Blowing-sand and sandstorm indicated the weather phenomenon in which the wind lifts the dust from the ground, making the horizontal visibility in the range of 1–10 km and less than 1 km, respectively.

In spring, the selected clear days were March 1, April 25, April 30, and May 12; the floating-dust days were March 6, March 7, March 9, March 16, and March 20; the blowing-sand days were March 17, March 29, April 5, April 12, and April 19; and the sandstorm days were March 3, March 19, April 2, April 11, and May 11. Their averages under these different weather conditions were then calculated.

In summer, the selected typical clear days were June 19, July 22, July 23, and August 21; the floating-dust days were June 1 and 2; the blowing-sand days were June 5, June 15, June 28, July 16, and July 24; and the sandstorm days were June 16, June 30, July 15, August 23, and August 29. Then, their averages were calculated under these different weather conditions.

Results and Discussion

Monthly variation of total, direct, and scattered radiations

In 2018, the total, direct, and scattered radiations at Xiaotang show fluctuating distributions with an evident seasonality (Fig. 3). The annual total, direct, and scattered radiations are 5781.8, 2337.9, and 3323.8 MJ m−2, respectively. The total annual radiation at the Xiaotang area (5781.8 MJ m−2) is lower than that at the Tazhong Station (6515 MJ m−2), which is in the hinterland of the Taklimakan Desert.

Figure 3 Annual variations in the total, direct, and scattered radiations at the Xiaotang area.

Figure 4 shows that the maximum monthly total radiation concentration occurs from June to August and the total radiation peaks in July (679.8 MJ m−2), accounting for 11.7% of the total radiation for the whole year.

Figure 4 Monthly variation of total radiation, direct radiation, and scattered radiation at Xiaotang.

The total annual direct radiation is 2337.9 MJ m−2, accounting for 42.4% of the annual total radiation. The peak value of the monthly total amount is seen in August (317.3 MJ m−2), accounting for ∼12.6% of the total annual direct radiation, and it reaches the lowest value (102.1 MJ m−2) in December (winter).

The scattered radiation begins to rise in January (138.1 MJ m−2) and reaches the peak value (455.7 MJ m−2) in May, which accounts for 13.7% of the total annual scattered radiation. Under high solar height angles observed in June, July, and August, the scattered radiation is 396.4, 353.4, and 314.9 MJ m−2, respectively. The total, direct, and scattered radiations reach the lowest levels in December, with monthly totals of 242.4, 102.1, and 133.6 MJ m−2, respectively.

Figure 5 shows that high daily variations of solar radiation reflect frequent changes in weather conditions and synoptic events. The amount of total radiation gradually increases from spring, and the peak value is reached in summer (1967.8 MJ m−2), which accounts for 34.03% of the annual total radiation; the values are mainly concentrated between 15 and 30 MJ m−2. It reaches the lowest value in winter (862.9 MJ m−2). The values in December are mainly concentrated between 7 and 9 MJ m−2 for 23 days, and the values in January–February are mainly concentrated between 9 and 12 MJ m−2 for 31 days. The amount of total radiation follows an order of summer > spring > autumn > winter.

Figure 5 Daily variation of total radiation, direct radiation, and scattered radiation at Xiaotang in 2018.

The amount of direct radiation is less in spring (449.5 MJ m−2). The total amount of direct radiation in summer is 864.4 MJ m−2, accounting for 35.3% of the total annual direct radiation. The amounts in spring and summer are mainly concentrated between 0 and 15 MJ m−2, with large fluctuations and lasting for 164 days. The value gradually decreases in autumn. In winter, the amount of direct radiation reaches the lowest value of 364.3 MJ m−2, with the magnitude mainly concentrated between 2 and 5 MJ m−2 for 54 days. The amount of direct radiation follows an order of summer > autumn > spring > winter.

During the year, the amount of scattered radiation reaches the maximum value in spring (1182.4 MJ m−2). The amount of scattered radiation in summer is 1064.7 MJ m−2, and the total amount in spring and summer accounts for 67.61% of the total scattered radiation in the year. The amounts of scattered radiation in the two seasons are mainly concentrated between 8 and 18 MJ m−2, with great fluctuations and lasting for 158 days. In spring and summer, the scattered radiation lasts for 141 days longer than the direct radiation. The scattering radiation in autumn is 619.1 MJ m−2, and the value gradually decreases. The lowest value of scattered radiation is 457.7 MJ m−2 (winter), accounting for 13.8% of the annual total amount of scattered radiation. The value of scattered radiation is mainly concentrated between 3 and 6 MJ m−2 and lasts for 80 days in December. The amount of scattered radiation follows an order of spring > summer > autumn > winter.

Relationship between solar radiation and solar height angle

Linear regression is used to analyze the change in total radiation with the solar height angle under different weather conditions (Fig. 6). The square values of the correlation coefficients are 0.97 and 0.81 for clear and sandstorm days, respectively. In sandstorm weather, the total radiation slowly increases with the increase in the solar height angle, mainly due to the increase in dust content in the air. When the solar height angle is 10°, the total radiation amounts in clear, floating-dust, blowing-sand, and sandstorm days are 118.6, 19.7, 10.4, and 4.0 W m−2, respectively. The higher the solar height angle, the more the radiation received by the ground. However, at the same solar height angle, the total radiation is the highest on clear days, while the total radiation in sandstorm weather is the lowest (Fig. 6). Solar radiation is affected by the solar height angle, atmospheric conditions, latitude, water vapor, cloud cover, and other factors. Here, this is mainly the influence of atmospheric conditions on solar radiation.

Characteristics of atmospheric transmittance

In total, 55.9% of astronomical radiation enters the ground, and the annual variation range is mainly between 30% and 74%. On clear days, the total radiation reaching the ground is 68.1%, with a variation range of 60%–70%. The total radiation reaching the ground on dusty days is 47.6%, with a variation range of 30%–60%. The variation range on clear days is mainly concentrated between 60% and 70% and that on the sandy and dusty days is mainly concentrated between 30% and 60%. The annual average atmospheric transmittances on clear, floating-dust, blowing-sand, and sandstorm days at the Xiaotang area are 0.67, 0.46, 0.44, and 0.30, respectively. Many studies show that the particle-size distribution, vertical distribution, and mineral composition of dust have important effects on radiation (Tegen & Fung, 1994; Sokolik & Golitsyn, 1993a; Sokolik & Golitsyn, 1993b; Sokolik & Toon, 1996).

As seen in Fig. 7, atmospheric conditions affect the total and scattered radiations. The higher the atmospheric transmittance, the weaker the scattered radiation and the smaller the ratio of the scattered radiation to the total radiation (Rd/Rs) (Fig. 7A). By contrast, as the atmospheric transmittance decreases, the scattered radiation and Rd/Rs increase. The ratio of the scattering radiation to the astronomical radiation (Rd/Qd) indicates the contribution of the atmosphere to scattered radiation (relative to astronomical radiation). (Fig. 7B). However, when the atmospheric transmittance is larger than 0.5, Rd/Qd begins to decrease.

Figure 6 Relationship between total radiation and solar height angle on clear, floating-dust, blowing-sand, and sandstorm days at the Xiaotang area.

Figure 7 (A) Relationship of atmospheric transmittance and ratio of scattered radiation to total radiation (Rd/Rs); (B) relationship of atmospheric transmittance and ratio of scattered radiation to astronomical radiation (Rd/Rs).

Diurnal variation of radiation on dusty day in spring and summer

Figure 8 shows that the distributions of hourly total, direct, and scattered radiations are normal during midday on clear days. The amount of radiation increases from sunrise and reaches the maximum value at 12:00 LST or 13:00 LST in spring and summer. By contrast, the distribution becomes irregular owing to the reduction in total radiation and direct radiation associated with dust aerosols. The value of direct radiation decreases at the Xiaotang Station. In spring (Fig. 8A), the daily peaks of total, direct, and scattered radiations on clear days are 862.4, 504.5, and 339.6 W m−2, while in summer (Fig. 8B), the daily peaks are 915.8, 643.4, and 253.5 W m−2, respectively.

Figure 8 Diurnal variation of total radiation, direct radiation, and scattered radiation on clear, floating-dust, blowing-sand, and sandstorm days in spring (A) and summer (B) at Xiaotang.

In spring (Fig. 8A), compared with the radiation values on clear days, the daily total values of total radiation decrease by 41.7%, 43.8%, and 62.8%; those of direct radiation decrease by 89.6%, 88.0%, and 97.6%; and the total daily values of scattered radiation increase by 9.1%, 6.4%, and 4.9% in floating-dust, blowing-sand, and sandstorm days, respectively.

Figure 8A shows that the daily variation curves of total and scattered radiations on blowing-sand and sandstorm days are similar, and the magnitudes are close. In summer (Fig. 8B), compared with the radiation values on clear days, the daily total values of total radiation decrease by 18.5%, 26.9%, and 57.5%, in floating-dust, blowing-sand, and sandstorm days, respectively. Those of direct radiation decrease by 81.5%, 76.4%, and 89.3%; and the total daily values of scattered radiation increase by 52.2%, 51.1%, and 9.2% in floating-dust, blowing-sand, and sandstorm days, respectively.

In general, the impact of sand and dust on radiation at the Xiaotang area is greater in spring than that in summer. According to the ground observation records (Table 2), the rainfall at the Xiaotang area is mainly concentrated in summer, with a total of 37 days. The occurrence of sandstorms is often accompanied by rainfall, which reduces the dust content in the air. In summer, there are nine sandstorms, five of which are accompanied by rainfall. In spring, there are more dusty days and less rainfall. Wang, Dong & Chen (2001) calculated the sand transport rate formula fitted using field-measured data and found that the annual sediment transport in the Xiaotang area was 3,800 kg m−1, which is higher than the annual sediment transport in some oases on the edge of Tarim (Yang, Li & He, 2012), and the direction of sand transport was relatively scattered in the Xiaotang area. The underlying soil comprises fine (125–250 µm) and very fine (62.5–125 µm) sand. The smaller the particle size, the stronger the scattering effect on light and the larger the scattering angle.

Table 2 Monthly number of days of dust days occurrence at Xiaotang in 2018.

Season	Month	Floating dust	Blowing sand	Sandstorm	Dust devil	Rainfall	
Spring	March	5	3	3	0	0	
April	4	5	4	2	4	
May	7	3	8	7	5	
Summer	June	2	6	3	4	10	
July	0	8	1	4	16	
August	0	4	5	7	11	

Probability distribution characteristics of total, direct, and scattered radiations in different dust weathers

The total, direct, and scattered radiations at the Xiaotang area differed under the various weather conditions. Therefore, the distribution law was further analyzed, and the total, direct, and scattered radiations on clear, floating-dust, blowing-sand, and sandstorm days at the Xiaotang area were statistically examined. The probability distribution of the radiations is shown in Fig. 9.

Figure 9 Probability distribution of total, direct, and scattered radiations values on clear, floating-dust, blowing-sand, and sandstorm days at the Xiaotang area.

The total radiation can reach 900–100 W m−2 in clear, floating-dust, and blowing-sand weather, while the value varies between 700 and 800 W m−2on sandstorm days. In sandstorm weather, the probability of total radiation >600 W m−2 is only 2.2%. On clear, floating-dust, blowing-sand, and sandstorm days, the magnitude in high-value of total radiation areas gradually decreases, gradually concentrating in low-value of total radiation areas. The weakening of total radiation by dust is mainly concentrated in high-value of total radiation areas.

The maximum value of direct radiation varies from 800 to 900 W m−2 on clear days, while the value varies between 100 and 200 W m−2in sandstorm days. The probabilities of the occurrence of direct radiation <500 W m−2 are 89.6%, 100%, 99.4%, and 100% on clear, floating-dust, blowing-sand, and sandstorm days, respectively. Compared with total radiation, sand and dust have a stronger weakening effect on direct radiation; especially in sandstorm weather, the magnitude is below 200 W m−2. The probabilities of occurrence of direct radiation <200W m−2 are 44.5%, 93.5%, 91.3%, and 100% in clear, floating-dust, blowing-sand, and sandstorm days, respectively.

On clear days, scattered radiation is mainly concentrated between 1 and 300 W m−2. In sandy and dusty weather, the maximum scattered radiation can exceed 600 W m−2. The probabilities of scattered radiation <300 W m−2 occurring on clear, floating-dust, blowing-sand, and sandstorm days are 94.2%, 62.5%, 61.1%, and 72.1% and those for the occurrence of scattered radiation <600 W m−2 are 100%, 99.1%, 98.1%, and 100%, respectively. Therefore, as the content of sand and dust in the air increases within a certain range, scattered radiation can be enhanced, especially in floating-dust and blowing-sand weather.

Characteristics of atmospheric transparency coefficient

The average P2 values on clear, floating-dust, blowing-sand, and sandstorm days at the Xiaotang area are 0.57, 0.34, 0.17, and 0.08, respectively. The total, direct, and scattered radiations are closely related to P2 (Fig. 10). P2 has a positive correlation with the ratio of the direct radiation to the total radiation (Rb/Rs) (R2 = 0.78) (Fig. 10A). Rb/Rs increases with an increase in P 2, showing an asymmetrical relationship. When P2 = 0, Rb/Rs tends to 0. P 2 is inversely correlated with Rd/Rs (R2 = 0.49) (Fig. 10B); Rd/Rs decreases with an increase in P 2. When P2 = 0, Rd/R s tends to 1.

Figure 10 (A) Relationship between atmospheric transparency coefficient and the ratio of the direct radiation to the total radiation; (B) relationship between atmospheric transparency coefficient and the ratio of the scattered radiation to the total radiation.

Direct radiation, scattered radiation, and AOD

Figure 11 displays the temporal changes in direct radiation, scattered radiation, and AOD in 2018. The figure shows that the annual variation of AOD is almost the same as that of scattered radiation, reaching the maximum in spring, May (0.71). The two variations have a good consistency. AOD is negatively correlated with direct radiation; there is an evident positive correlation between AOD and scattered radiation, while AOD is the most influencing and important factor for scattered radiation at the Xiaotang area (Huang et al., 2020). AOD is negatively correlated with direct and scattered radiations. Because the Xiaotang area is in the northern margin of Taklimakan Desert, the main aerosol in the air is dust, which leads to the enhancement of scattered radiation.

Figure 11 Monthly variation of direct radiation, scattered radiation, and AOD at Xiaotang in 2018.

Discussion

Analysis of the radiation data of the Xiaotang area in the northern margin of the Taklimakan Desert revealed that the radiation in this area was significantly different from that at the Hade Station (Taklimakan Desert) (Jin et al., 2014), Tazhong Station (Taklimakan Desert hinterland) (Mamtimin et al., 2014), other desert sites (Yang et al., 2018; Gao et al., 2021), high-latitude areas (Qiao, Gu & Tang, 2008; Zhang et al., 2015; Zhang, Zhang & Wang, 2013), and other areas (Qian et al., 2011; Qiu, 1996).

The annual total radiation was 5781.8 MJ m−2 at the Xiaotang area in 2018. The annual total radiation was 6515.0 MJ m−2 at the Tazhong Station during 2007–2011. The annual total radiation was 5774.5 MJ m−2 at the Hade Station in 2011. The annual total radiation was 6191.19 MJ m−2 at the Kelameili Station (Gurbantünggüt Desert) in 2017. The annual total radiation was 4635.0 MJ m−2 at the Heihe River Basin The annual total radiation was 5988 MJ m−2 at Hexi corridor. The total annual radiation was >6000 MJ m−2 at the Golmud area during 1993–2011.

The annual total amount of direct radiation was 2337.9 MJ m−2 at the Xiaotang area in 2018. The annual total amount of direct radiation was 2203.5 MJ m−2 at the Tazhong Station during 2007–2011; The annual total amount of direct radiation was 2772.6 MJ m−2 at the Heihe River Basin.

The annual total amount of scattered radiation was 3323.8 MJ m−2 at the Xiaotang area in 2018. The annual total amount of scattered radiation was 3628.5 MJ m−2 at the Tazhong Station during 2007–2011. The annual total amount of scattered radiation was 1682.6 MJ m−2 at Heihe River Basin.

The main reason for this difference is that the Xiaotang area lies in the desert–oasis transition zone in the northern margin of the Taklimakan Desert. The sandy and dusty days are as many as 30 in spring and summer. Sand and dust are easily brought into the air owing to the lack of vegetation blockage. As the sand–dust content in the air increases and the transparency of the atmosphere decreases, the radiation received by the ground decreases. Precipitation, which plays a major role in eliminating sand and dust from the summer air, is mainly concentrated in summer in the Xiaotang area. The maximum monthly values of total radiation at the Hade Station and Xiaotang area are close; there are little differences in latitude, altitude, and natural environment between these two places.

Tazhong Station has a lower geographical latitude (38°58′N, 83°39′E; altitude = 1,090 m) than the other desert stations (Hade, Xiaotang, Kelameili, and Guaizihu Station). Influenced by the solar altitude angle, the radiation received by the ground at the Tazhong Station is greater the other desert stations. As a mobile desert, the Taklimakan Desert has more sandy and dusty days in spring and summer. In addition, the main particle sizes comprise those of dust, sand, and fine sand (62.5–125 µm); they are light and stay in the air for a long time.

Gurbantünggüt Desert is a fixed and semi-fixed desert with a high latitude and low altitude (45°14′N, 87°35′E; altitude = 531 m); its annual total radiation amount is larger than those at the Xiaotang and Hade Stations, which have low latitudes. The main reason is that in spring and summer, Kelameili Station plays a key role in fixing sand on the soil surface layer owing to the vigorous growth of vegetation and the soil at the observation point has a certain water storage capacity; thus, sand–dust weather cannot easily occur, and sandy and dusty days are few (Gao et al., 2021).

In the Guaizihu Station (41°22′N, 102°22′E; altitude = 960 m), as shown by statistics of weather phenomena, the number of dusty days in spring and summer is considerably smaller than that at the Tazhong Station (Yang et al., 2018).

The stations in the Qinghai–Tibet Plateau and Golmud area have higher altitudes (3250 and 2807.6 m) and less scattered radiation. When solar rays reach the ground, they pass through a thin atmosphere and the scattering effect is weak. The maximum scattering radiation in the Qinghai–Tibet Plateau, Golmud area, and Xiaotang area occurs in spring, mainly because June–September is the rainfall season in the three areas. Vegetation grows and propagates in this season, and the relative humidity of the air is high; thus, the value of scattering radiation is higher than the value of direct radiation in spring (Zhang et al., 2015).

Conclusion

(1). In 2018, the Xiaotang area’s total, direct, and scattered radiations showed fluctuating distributions (5781.8, 2337.9, and 3323.8 MJ m−2, respectively). The total and direct radiations reached peak values in July (679.8 and 310.0 MJ m−2, respectively). The total amount of direct radiation was highest in August (317.3 MJ m−2); the total amount of scattered radiation reached the peak value in May (455.7 MJ m−2). The three factors, i.e., total, direct, and scattered radiations, were observed to be the lowest in December.

(2). In 2018, the Xiaotang area’s total, direct, and scattered radiations showed an evident seasonality. There were many windy and sandy days in spring and summer; as the content of sand and dust in the air increased, the scattered radiation increased, with the highest value occurring in spring. Summer rainfall played a role in cleaning the air; consequently, the atmospheric transparency increased and the total and direct radiations became the largest in summer. Therefore, the weakening of the effect of dust on radiation at the Xiaotang area was greater in spring than that in summer. In winter, the solar altitude angle and total, direct, and scattered radiations were the lowest.

(3). The annual average atmospheric transmittances on clear and dusty days at the Xiaotang area were 0.67 and 0.46, respectively. Dust had the greatest influence on direct radiation, making it concentrate in low-value areas (1-200W m−2). The probability of total radiation <200 W m−2 occurring in sandstorm weather was 100%. Sand and dust can enhance the scattering radiation in a certain range; if the sand and dust content in the air is too high (atmospheric transmittance was larger than 0.5), the scattering radiation will be weakened.

(4). Total, direct, and scattered radiations are closely related to P 2. At the Xiaotang area, P 2 value was the lowest in spring and summer but the highest in September–December. On clear, floating-dust, blowing-sand, and sandstorm days, P2 values were 0.57, 0.34, 0.17, and 0.08, respectively.

Supplemental Information

Supplemental Information 1 Raw Data

2018 solar radiation data from Xiaotang Station in the Taklimakan Desert.

Click here for additional data file.

The authors would like to thank the Institute of Desert Meteorology, China Meteorological Administration, for providing radiation and meteorological data.

Additional Information and Declarations

Competing Interests

Author Contributions

Data Availability

The authors declare there are no competing interests.

Lili Jin conceived and designed the experiments, authored or reviewed drafts of the paper, and approved the final draft.

Sasa Zhou conceived and designed the experiments, analyzed the data, prepared figures and/or tables, and approved the final draft.

Qing He conceived and designed the experiments, performed the experiments, authored or reviewed drafts of the paper, and approved the final draft.

Alim Abbas analyzed the data, prepared figures and/or tables, authored or reviewed drafts of the paper, and approved the final draft.

The following information was supplied regarding data availability:

This data is available in the Supplementary File.

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
