# Peer review of "Characteristics of solar radiation at Xiaotang, in the northern marginal zone of the Taklimakan Desert"

_PeerJ, doi:10.7717/peerj.12373_

## Round 0.1 · original submission · Major Revisions

I have completed my evaluation of your manuscript. One reviewer recommend reconsideration of your manuscript following minor revision. However, the other pointed out many problems and recommend rejection. Therefore, I invite you to resubmit your manuscript after addressing the comments below.

Reviewer 1 ·

Basic reporting

no comment

Experimental design

no comment

Validity of the findings

no comment

Additional comments

The characteristics of solar radiation and the influence of sand and dust on solar radiation in the northern margin of Taklimakan Desert were analyzed using radiation observation data in 2018. The results showed that the annual total, direct, and scattered radiation at Xiaotang were 5781.8, 2337.9, and 3323.8 MJ·m−2, respectively. The aerosol optical depth corresponded well with the scattered radiation. Under the same solar height angle, total radiation was higher during clear days but lower on sandstorm days. The average atmospheric transmittance was 0.67 on a clear day, and 0.46 on sand-and-dust days. When the atmospheric transmittance was 0.5, the increase in scattering radiation by aerosol in the air began to decrease. Probability analysis of radiation indicated the following probabilities of total radiation <500 W·m−2 occurring on clear, floating-dust, blowing-sand, and sandstorm days: 67.1%, 76.3%, 76.1%, and 91.8%, respectively. Dust had the greatest influence on direct radiation; the probabilities of direct radiation <200 W·m−2 occurring on clear, floating-dust, blowing-sand, and sandstorm days were 44.5%, 93.5%, 91.3%, and 100%, respectively, whereas those of scattered radiation <600W·m−2 were 100%, 99.1%, 98.1%, and 100%, respectively.

Based on the analysis of observational data, this study is useful to understand solar radiation transfer at the Taklimakan Desert and can be accepted after revision. The comments and suggestions are as follows:

Line 141, “The clock of the collector adopts the local real solar time, which differs from Beijing time by 2 h 22 min 48 s”, it is better to show the difference specifically, i.e., earlier or later.

Line 150, “Due to the high sensitivity of the instrument, it outputs in the form of negative numbers”, it is not clear, please explain the reasons for the negative numbers.

Line 164, “Rb’is the direct radiation on the vertical plane” should be corrected. It is really on the vertical plane?

Lines 170-172, “The total solar astronomical radiation incident on the top of the atmosphere is absorbed and scattered by atmospheric molecules and aerosols, and the radiation received by the ground is the total radiation” should be rewritten.

Lines, 243, 250, 261, What is meaning for “Summer > spring > autumn > winter” ?

Line 620, “The value of r is mainly concentrated between 3 and 6 MJ·m−2 and lasts 80 days”, what is r?

Lines 301, 470, “When the atmospheric transmittance is 0.5, the ratio of scattered radiation to astronomical radiation (Rd/Qd) begins to decrease”, does it mean When the atmospheric transmittance is larger than 0.5? please explain this sentence more clearly.

Line 304, “Figure 8 shows that the distributions of hourly total radiation, direct radiation, and scattered radiation are symmetric during midday on clear days”, should be rewritten.

Line 386, “AOD is negatively correlated with DSSRSBD”, please explain DSSRSBD.

Line 458, please remove “the atmospheric transmittance increased”.

Lines 471-473, “Sand and dust can enhance the scattering radiation in a certain range; if the sand and dust content in the air is too high, the scattering radiation will be weakened”, it is better to give the specific conditions for the sand and dust content in the air is too high.

“Figure 9 Statistics of total radiation, direct radiation, and scattered radiation on clear, floating-dust, blowing-sand, and sandstorm da ys at Xiaotang”, please explain da, and ys.

Reviewer 2 ·

Basic reporting

no comment

Experimental design

no comment

Validity of the findings

no comment

Additional comments

The manuscript reported radiation data for 2018 observed around the north margin of the Taklimakan desert, and discussed the diurnal and annual variation, the relations with solar height angle, aerosol load, and the atmospheric transmittance and the transparency coefficient. However, the observed data were analyzed at a superficial level, from which it is hard to gain new scientific knowledge beyond the common sense or insight about the atmospheric characteristics of that area.

Besides, there are many errors in the statements unacceptable for publication:

Line 145: the authors said, "... the radiometer is wiped before sunrise. However, the radiometer is not wiped during sand and dust weather to capture the impact of sand and dust on radiation." That will lead to lower values of the radiation during the sand and dust days.

Line 183-187: What is the definition of the "air (or atmospheric) quality"? Should it be "air mass"? Why is It 2 when the solar altitude angle is 90 degree? The bibliography Hu et al. (2008) is missing, Hu et al. (2015) seems not accessible for english readers. If there is not any source in English, the author should have talked more about equation (5).

Line 269: the authors said "When the solar height angle is less than 10°, the total radiation amounts in clear, floating-dust, blowing-sand, and sandstorm days are similar to the fitting curve of the solar height angle." The similarity can not be seen in the figure. If the similarity means the smaller absolute bias, that is meaningless.

Line 276: the authors said, "As the solar height angle increases, the total radiation becomes more affected by the weather." Similarly to the last comment, that can not be seen in the figure.
Line 282: The title should be about the atmospheric transmittance.

Line 290: The "sand and dust" day seems not defined yet.

Line 320: the authors said, "In summer (Fig. 8(B)), the daily total value of total radiation decreases by 13.2%, 22.7%, and 51.5% compared with those on clear, floating-dust, blowing-sand, and sandstorm days in spring, respectively." It seem one value is missing here, and I can not see the values in summer are smaller (if not bigger) that those in spring.

Line 326-328: It can be seen from table 2, "36 days" should be "37 days". "eight sandstorms" should be "nine sandstorms".

Line 320: The source of the the total sediment transport value should be provided.

Line 381: What is the source of AOD data?

Line 387: what is DSSR_SBD ?

In section discussion, it is not proper to generally state the annual total radiation at Xiaotang is higher or lower than those at other sites, because there is only one year of observation at Xiaotang, and the data at the sites are for different periods (years), which can be seen from the publication dates.

Line 197: “A total cloudiness of >20% and an inverted-U-shaped daily change in total radiation were taken to indicate a clear day”, where > should be <.

Line 200-203: The "blowing sand" were defined twice.

Some statements are hard to understand, maybe due to language error, like:

Line 96: "the attenuation of dust aerosols and solar radiation is as high as 40% in the summer."

Line 184: "To overcome the Forbes effect, that is, the atmospheric quality interfering with changes in solar spectral composition (Hu et al., 2015)."

Line 466: The weakening of total radiation dust aerosol was mainly concentrated in high-value areas.

---

## Round 0.2 · Minor Revisions

I confirm that the criticisms from reviewer 2 have been addressed.. Please revise the manuscript based on the comments from reviewer 1.

Reviewer 1 ·

Basic reporting

no comment

Experimental design

no comment

Validity of the findings

no comment

Additional comments

The authors revised the manuscript, minor revisions are required.

L144, please add the model of A CR3000 micrologger (Campbell, United States).

L313, “value of direct radiation decreases sharply in at the Xiaotang Station” should be corrected.

---

## Round 0.3 · accepted · Accept

The current version is ready for publication. Thank you for responding positively to the comments of the reviewers and for your patience in going through the review process. We appreciate your support of the journal and hope you will publish future work with PeerJ.